# New Insight into Short Time Exogenous Formaldehyde Application Mediated Changes in *Chlorophytum comosum* L. (Spider Plant) Cellular Metabolism

**DOI:** 10.3390/cells12020232

**Published:** 2023-01-05

**Authors:** Maria Skłodowska, Urszula Świercz-Pietrasiak, Małgorzata Krasoń, Anita Chuderska, Justyna Nawrocka

**Affiliations:** Department of Plant Physiology and Biochemistry, Faculty of Biology and Environmental Protection, University of Lodz, 90-237 Lodz, Poland

**Keywords:** *Chlorophytum comosum* L., exogenous formaldehyde, glucose, sucrose, alanine and aspartate aminotransferases, photosynthetic pigments, phenolics

## Abstract

*Chlorophytum comosum* L. plants are known to effectively absorb air pollutants, including formaldehyde (HCHO). Since the metabolic and defense responses of *C. comosum* to HCHO are poorly understood, in the present study, biochemical changes in *C. comosum* leaves induced by 48 h exposure to exogenous HCHO, applied as 20 mg m^−3^, were analyzed. The observed changes showed that HCHO treatment caused no visible harmful effects on *C. comosum* leaves and seemed to be effectively metabolized by this plant. HCHO application caused no changes in total chlorophyll (Chl) and Chl a content, increased Chl a/b ratio, and decreased Chl b and carotenoid content. HCHO treatment affected sugar metabolism, towards the utilization of sucrose and synthesis or accumulation of glucose, and decreased activities of aspartate and alanine aminotransferases, suggesting that these enzymes do not play any pivotal role in amino acid transformations during HCHO assimilation. The total phenolic content in leaf tissues did not change in comparison to the untreated plants. The obtained results suggest that HCHO affects nitrogen and carbohydrate metabolism, effectively influencing photosynthesis, shortly after plant exposure to this volatile compound. It may be suggested that the observed changes are related to early HCHO stress symptoms or an early step of the adaptation of cells to HCHO treatment. The presented results confirm for the first time the direct influence of short time HCHO exposure on the studied parameters in the *C. comosum* plant leaf tissues.

## 1. Introduction

Among air pollutants, formaldehyde (HCHO) is considered one of the most common and hazardous [1]. Because of the constant release of HCHO from synthetic materials intensively used in the construction and furnishing of new buildings, the indoor concentration of this pollutant more and more often exceeds the permissible norms [2,3]. Endogenous HCHO present in plant cells at low concentrations, around 0.1–10 μmol/g fresh weight (FW), is a natural intermediate that arises in plant one-carbon (C1) metabolism [4,5]. Additionally, plants can absorb from the air and metabolize exogenous HCHO, however, they show different tolerance levels to this volatile organic compound. For certain plants, HCHO is toxic, even at low levels [6], especially when the exposure to a stress factor is prolonged over time. For example, a significant 50% decrease in stomatal conductance in *Ficus benjamina* after exposure for several weeks to 0.05 ppm HCHO was shown [7], and yellowing and necrotic symptoms on leaves of Arabidopsis, tobacco, and geranium treated with HCHO were presented [8]. On the other hand, over 60 plant species seem to be particularly effective in removing formaldehyde from the air under various conditions [9,10]. Therefore, formaldehyde phytoremediation has been paid a great deal of attention in the past decade [9,11]. 

It was shown that *Chlorophytum comosum,* known as a spider plant, can take up and extensively metabolize gaseous HCHO as a carbon source [12,13]. Most of the formaldehyde taken up from the air by spider plants is accumulated in shoots, some is transported to roots and released into the rhizosphere solution, and some is volatilized into the air again. Older leaves show a greater ability to remove formaldehyde from the air than mature big leaves [13]. Compared to other plant species, *C. comosum* shows a relatively high tolerance to formaldehyde. The plant was included in a group of plants with strong formaldehyde absorption capacity, oscillating around 8 mg m^−2^ of leaf area, while 90% of the plants absorb less than 5 mg m^−2^ [11]. Moreover, *C. comosum* was recommended to be used for formaldehyde purification, even at high concentrations such as 15 mg m^−3^ [14]. Although the formaldehyde removing plants have been intensively studied under varied environmental conditions, including different HCHO concentrations and times of exposure, indoor temperature, relative humidity, and illumination, the physiological and biochemical bases of HCHO phytoremediation and metabolism, as well as plant defense responses to this compound have not been fully elucidated [8,15,16]. Until now, it has been established that in plants, formaldehyde is mainly metabolized in the cells of leaf palisade and sponge tissues, in the mesophyll, and compounds and enzymes which are the first to bind and convert formaldehyde are localized mainly in the cytoplasm [17]. The main mechanism of formaldehyde loss in plants is its decomposition in plant tissue caused by enzyme and redox reaction [18]. It is suggested that exogenous HCHO entering plant cells is first oxidized in the cytosol [19,20]. During the enzymatic oxidation, HCHO may be spontaneously coupled with glutathione (GSH), subsequently converted to S-formylglutathione (FGSH), and then hydrolyzed to formate (HCOOH) [9,10]. Subsequently, generated HCOOH may be oxidized to CO_2_ through a reaction catalyzed by HCOOH dehydrogenase (FDH) or it may be transported to the other organelles, including chloroplasts and mitochondria, where it enters different pathways, for example, Calvin cycle and Krebs cycle, respectively [4,21,22,23]. Alternatively, rather than oxidation to CO_2_, HCHO in plant cells can be quickly condensed with tetrahydrofolate (THF) to 5,10-CH_2_-THF and enter C1 metabolism in the mitochondria where it can be used for amino acid synthesis [10]. The evidence implies that the Calvin cycle and C1 metabolism may function simultaneously during HCHO metabolism and detoxification in Arabidopsis [10]. Further information regarding the initial steps of HCHO metabolism is included in Appendix A. 

It was established that carbon delivered by HCHO might be incorporated into organic acids, amino acids, free sugars, and lipids as well as cell wall components [12]. Chloroplasts are considered as important organelles which may participate in formaldehyde metabolism. It is suggested that after oxidation of HCOOH to CO_2_ in chloroplasts it later may be directly assimilated by the Calvin cycle into sugars [24]. The HCHO-derived ^13^C- or ^14^C-labeled carbon incorporated in glucose (Gluc), fructose (Fruc) and sucrose (Suc) was detected in several studies [7,25,26]. Regarding the quantitative analyses of sugars in plants treated with HCHO, it is suggested that at the beginning of the treatment sugars are used as an energy source to plant defense response to formaldehyde, and then are regenerated with the HCHO-delivered carbon [25,26]. However, the quantitative analysis of the separated sugars, including Glu and Suc, the first metabolites originating from photosynthetic carbon dioxide fixation [27], in plants treated with HCHO, has been investigated only to a small extent. 

Amino acids are the large group of metabolites in which HCHO-derived carbon is incorporated. In different plants, HCHO was metabolized and converted to glutamate (Glu), glutamine (Gln), arginine (Arg), asparagine (Asn), serine (Ser), alanine (Ala), methionine (Met), and glycine (Gly) [25,26,28]. Even though there is considerable knowledge regarding the type of amino acids containing carbon derived from HCHO, the knowledge about their synthesis and transformations, as well as the enzymes involved in these pathways in plants treated with HCHO exogenously, is limited. For example, little is known about aspartate aminotransferase (AspAT, EC 2.6.1.1) involved in reversible transamination between Glu and oxaloacetate to generate aspartate (Asp), Asn and 2-oxoglutarate [29], as well as alanine aminotransferase (AlaAT, EC 2.6.1.2) which catalyzes the reversible conversion of pyruvate and Glu into Ala and oxoglutarate and links primary carbon metabolism with the synthesis of various amino acids [30,31].

Regarding defense responses of plants against HCHO, most research focuses on the effect of this compound on the functioning and stability of chlorophyll (Chl), one of the most important pigments in plant photosynthesis, determining photosynthetic capacity [32,33], as well as carotenoids (Car), the major substances protecting Chl from damage [34]. The pigments were mostly studied in plants showing symptoms of phytotoxic influence of formaldehyde, including chlorosis. The negative influence of formaldehyde, especially at high concentrations, on the total Chl, Chl b, and Car content, and inhibition of photosynthesis was presented in different plants, including *Arabidopsis thaliana* [35]. On the other hand, in *Pisum sativum* L., HCHO caused paradoxical changes regarding Chl content, which was decreased by HCHO at low concentrations and increased by high concentrations [36]. Therefore, a need to perform additional analyses that would explain the basis of such a diverse influence of HCHO on photosynthetic pigments, should be performed. 

Phenolics are a large and diversified group of plant secondary metabolites such as phenolic acids, simple and complex flavonoids, and anthocyanins [37]. They play an important role in many processes, including defense responses in a plant [37,38]. Regarding plant response to HCHO, HCHO-derived ^14^C was incorporated into lignins in *Glycine max* L. suspension cells [20]. Moreover, HCHO was shown to enhance the expression of enzymes that are involved in phenolic transformations, such as peroxidase (POD) [11]. Such results suggest that phenolics may play some role in the defense response of plants to exogenous HCHO, however no studies exploring this role have been performed so far. 

Taking into account the current state of knowledge, mentioned above, the aim of this study was to extend the information on the metabolism of HCHO by *C. comosum*, and its defense responses to this compound when it is applied exogenously. The analyses included determination of sugar synthesis and transformation, as well as the analysis of content and functionality of Chl and Car. Among the novel issues raised, the analyses of AspAT and AlaAT activities as well as phenolic content should be mentioned. These parameters were not studied extensively in the context of HCHO detoxification. The characteristics of biochemical responses related to the mentioned parameters may be an important step in understanding the mechanisms of formaldehyde removal, allowing for more effective use of plants in phytoremediation of this harmful compound from the air.

## 2. Materials and Methods

### 2.1. Treatment of C. comosum Plants with Formaldehyde 

Small uniform-sized seedlings of *C. comosum* L. were separated from their mother plant and planted into sterilized (to eliminate microorganisms) peat-based substrate (commercial) in plastic pots (400 cm^3^), one seedling per pot. According to the manufacturer’s information, mineral nutrient content in 1 L of peat-based substrate was as follows: 80–120 mg N, 60–80 mg P, 160–220 mg K, 70–120 mg Mg and 1500–2000 mg Ca, with pH 5.5–6.0. The same commercially available substrate was used in other studies [39]. The plants were cultured under a controlled environment, in a growth room, at the temperature of 23 ± 0.5 °C with 16 h light/8 h dark photoperiod at a light intensity of 174 μmol∙m^−2^ s^−1^ photon flux density and 60–70% relative humidity. Additionally, once per week the plants were watered with a modified Hoagland nutrient solution (pH 6.4) containing 5 mM Ca(NO_3_)_2_ × 4H_2_O, 5 mM KNO_3_, 1 mM KH_2_PO_4_, 2 mM MgSO_4_ × 7H_2_O, 0.1 mM FeNaEDTA, 9.1 μM MnSO_4_ × 5H_2_O, 46.2 μM H_3_BO_4_, 0.32 μM CuSO_4_ × 5H_2_O, 7.7 μM ZnSO_4_ × 7H_2_O, and 0.54 μM Na_2_MoO_4_ × 2H_2_O [40]. The pH of the nutrient solution was adjusted to 6.4 using 0.1M KOH or HCl solution. Plants were grown for five months before the experiment. Five-month-old plants (about 30 cm high) were placed for two days in glass chambers with a volume of 0.22 m^3^ (60 cm length × 60 cm width × 60 cm height) which were made perfectly air-tight. A door was provided in front of the chamber which was sealed by adhesive foam-rubber insulation tape and adjustable metal clips. Two plants were placed in each chamber. All the chambers were localized in the growth room in the same, stable conditions, as mentioned above. The flowerpot and basin soil of each plant were covered with polyethylene film during the experiment.

General techniques for formaldehyde exposure were adopted from Su Y. and Liang Y. [13] modified method. Plants were divided into two groups. The first group, control plants, was exposed to the air without formaldehyde, dispersed by external pump-generated air circulation (6 L min^−1^). The second group, HCHO treatment, was exposed to gaseous formaldehyde at a concentration of 20 mg m^−3^ ± 0.04 mg m^−3^ dispersed by external pump into the air inside the box for 48 h. Formaldehyde at a concentration of 20 mg m^−3^ was prepared from 37% (*w/v*) formalin solution (Sigma Aldrich, St Louis, MO, USA). For this purpose, 50 µl of formalin solution was added to the vessel, which was distributed by air pump to the empty (without plants) box in a closed circuit. HCHO concentration was measured every 15 min until the concentration reached the stable level, and then after 12, 24 and 48 h (Appendix A).

Formaldehyde concentration was chosen based on our preliminary studies which showed it to be the threshold effective in the activation of visible biochemical changes in *C. comosum* plants. Regarding tested parameters, HCHO at lower concentrations did not cause any significant changes as compared to the control. Moreover, HCHO at the concentration of 20 mg m^−3^ showed no phytotoxic effects on the *C. comosum*. Similarly, HCHO at high concentrations, between 10 and 16 mg m^−3^, were used in studies with *C. comosum* and other plants [2,11].

### 2.2. Sample Preparation 

The HCHO concentration in the air of empty chambers and chambers with plants was controlled by collecting the air samples with an Aspirator Individual GiLAir pump for washers containing 2 cm^3^ of absorbent (50 mM phosphate buffer) and detected by acetylacetone spectrophotometry. Regarding chambers with plants, the air samples were collected at two time points: (i) just after the placement of plants into the chamber, and (ii) after 48 h of plant exposure to HCHO.

After 48 h of HCHO exposure, the leaves used for the biochemical analyses were taken from the middle of the plant, between the oldest and youngest leaves. The harvested leaves were washed with pre-chilled sterile water to remove free HCHO from the plant surface and dried with filter paper.

The fresh leaves were used for biochemical analyses of HCHO, Glu, Suc, photosynthetic pigment, and protein contents, as well as AspAT and AlaAT activities. Before the determination of phenolics, the leaf samples were briefly stored at −20 °C.

### 2.3. Determination of HCHO

For biochemical analysis of the formaldehyde concentration, 0.5 g samples of fresh leaves without main veins, were immediately homogenized in the frozen mortar in 5 cm^3^ ice-cold 0.05 M sodium phosphate buffer, pH 7.0, containing 1 mM EDTA and 1% polyvinylpyrrolidone, and centrifuged (20,000× *g*, 20 min, 4 °C). The content of HCHO in leaf tissues was determined with the spectrophotometric method of Su Y. and Liang Y. [13] by measuring the absorbance at the wavelength λ = 414 nm. The principle of the method is based on the reaction of HCHO with neutral solutions of acetylacetone and ammonium ions which leads to the formation of a yellow-colored complex of diacetyl dihydrolutidine. The samples contained 0.2 cm^3^ of the supernatant and 0.8 cm^3^ of a reagent, prepared by mixing 15 g of 2 M ammonium acetate and 0.3 cm^3^ of 50 mM acetic acid, and 0.2 cm^3^ of 20 mM acetylacetone in 100 cm^3^ of distilled water. Immediately after mixing the components, the reaction mixture was incubated at 60 °C for 15 min. The content of formaldehyde was calculated using the formaldehyde standard curve and presented as mg g^−1^ FW.

### 2.4. Determination of Gluc and Suc

Gluc and Suc contents were determined in the extracts obtained from leaf samples (0.5 g FW) after triple 80% ethanol extraction and centrifugation (20,000× *g*, 15 min). The ethanolic extracts were evaporated to dryness at 50 °C and the residue was resolubilized with distilled water. The contents of both sugars were assayed using the commercial enzymatic test (Boehringer Mannheim Biochemica, Mannheim, Germany) according to the manufacturer’s instruction and expressed in μg per mg protein determined in the extract used for enzyme activity analyses.

### 2.5. Determination of Enzyme Activities

AspAT and AlaAT activities were assayed spectrophotometrically in the same supernatants which were used for HCHO concentration analysis, as described by Gajewska et al. [41]. AspAT activity was assayed in the direction of the formation of oxaloacetate from aspartate by coupling the reaction to NADH oxidation by malate dehydrogenase. The reaction mixture (1.5 cm^3^) consisted of 0.1 M Tris–HCl buffer, pH 7.8, 5 mM EDTA, 0.2 M L-aspartic acid, 12 mM 2-oxoglutaric acid, 0.18 mM NADH, five units of malate dehydrogenase and enzyme extract. AlaAT activity was assayed in the direction of the formation of pyruvate from the alanine by coupling the reaction to NADH oxidation by lactate dehydrogenase. The reaction mixture (1.5 cm^3^) consisted of 0.1M Tris–HCl buffer pH 7.5, 0.5 M L-alanine, 15 mM 2-oxoglutaric acid, 0.18 mM NADH, five units of lactate dehydrogenase, and the enzyme extract. In the case of both AspAT and AlaAT, the reaction was conducted at 30 °C, and oxidation of NADH was monitored at 340 nm. AspAT and AlaAT activities were calculated using the absorption coefficient for NADH (ε = 6.22 mM^−1^ cm^−1^) and expressed in units, each representing the amount of enzyme catalyzing the formation of 1 nmol of product min^−1^ and expressed in U mg^−1^ protein. 

### 2.6. Determination of Photosynthetic Pigments

To determine the contents of Chl a, Chl b, and Car, the fresh leaves were extracted three times with 80% acetone (1:20 *w/v*) and centrifuged (33,000× *g*, 15 min). After measurement of the absorbance of the supernatants at 470 nm, 663.2 nm, and 646.8 nm, Car, Chl a, and Chl b contents, respectively, were calculated according to the method of Wellburn [42].

### 2.7. Chl Fluorescence Imaging

Visualization of Chl was performed according to Donaldson et al. 2018 [43] (imaging and spectroscopy of natural fluorophores in pine needles). Green fluorescent (GF) was followed on the upper side of the fresh leaf using confocal microscope (Leica TCS LSI macroscope with 5 × objective, 0.10 numerical aperture, Leica Microsystems). The 488 nm laser was used for the excitation of GF, and emission was measured in the window from 620 nm to 750 nm. LAS-AF Version 4.0.0 (Leica Microsystems, Wetzlar, Germany) was used to analyze the data. Fluorescence intensity was expressed in arbitrary units (AU). Calculations were performed for 10 randomly selected points.

### 2.8. Determination of Total Phenolics

Frozen leaf samples (1 g) were extracted twice with 15 cm^3^ of 80% methanol and after centrifugation (20,000× *g*, 20 min, 4 °C) the supernatant was used for measurement of the total phenolic content. 

The total phenolic content was determined using Folin–Ciocalteau reagent according to the Singleton and Rossi [44] method. The absorbance of the reaction product was measured at 725 nm and the phenolics content was expressed as milligrams per gram of FW, based on the calibration curve prepared for chlorogenic acid (Sigma Aldrich). 

### 2.9. Protein Determination 

Protein contents were measured in the same supernatants which were used for HCHO concentration analysis, as described by Bradford [45] using bovine serum albumin (BSA) as a standard.

### 2.10. Statistical Analysis

All the parameters were studied five times; therefore, each data point is the mean of five independent replicates (*n* = 5). Each replicate consisted of two to three plants. The significance of differences between mean values was determined by the nonparametric U Mann–Whitney Rank Sum Test using Statistica 13.1 software. Differences at *p* < 0.05 were considered significant. Data are given as mean values ± standard deviation. 

## 3. Results

### 3.1. Formaldehyde Concentration in the Air and in C. comosum 

The initial concentration of HCHO after placement of plants into the chamber remained stable, around 20 mg m^−3^ ± 0.04 mg m^−3^. After 48 h of plant exposure to HCHO, its concentration decreased to the range of 0.02 mg m^−3^ (Appendix A).

After 48 h of the exposure of plants to the gaseous HCHO no significant difference in HCHO concentration in the leaf tissues was observed compared to the control (Figure 1).

### 3.2. Gluc and Suc Contents

After 48 h of exposure of the *C. comosum* plants to gaseous formaldehyde, a significant enhancement of Gluc contents in the leaves was found (Figure 2).

The content of Gluc was increased by 63% as compared to the control, while that of Suc showed diminution by about 48% as compared to the control value.

### 3.3. AspAT and AlaAT Activities

The 48 h plant exposure to the gaseous formaldehyde significantly (*p* < 0.05) influenced the AspAT and AlaAT activities, which were lower than in the controls, by 20% and 40%, respectively (Figure 3).

### 3.4. Photosynthetic Pigment Contents

The data showed differentiation in changes of Chl a, Chl b, and Car contents in the leaves exposed for 48 h to gaseous formaldehyde applied as 20 mg m^−3^ (Table 1).

*C. comosum* leaves responded to gaseous formaldehyde exposure with a considerable decrease in Chl b and Car contents. The Chl b achieved only on average 78.6%, whereas Car was on average 63% of the values observed in the respective controls. The applied conditions did not significantly alter the Chl a content in the leaves after 48 h of exposure to HCHO (Table 1). The total Chl (a+b) content did not decrease in a significant manner, however the decreased Chl b content resulted in the change of the Chl a/b ratio value which was about 1.3-fold higher in the treated ones.

### 3.5. Intensity of Chl Fluorescence

Different intensity of Chl fluorescence as compared to the control was demonstrated in fresh *C. comosum* leaves treated for 48 h with gaseous formaldehyde at a dose of 20 mg m^−3^ (Figure 4 and Figure 5). In vivo observations of *C. comosum* leaves showed reduced emission of red fluorescence from Chl and significantly lower intensity of Chl fluorescence compared to the control.

### 3.6. Total Phenolic Content

There was no significant difference in total phenolic contents between the control and the plants exposed to the gaseous formaldehyde (Figure 6).

## 4. Discussion

The results of the present studies showed that *C. comosum* might effectively participate in the removal of HCHO from the air. Since the initial concentration of HCHO oscillated around 20 mg m^−3^, and after 48 h of plant exposure it decreased to 0.02 mg m^−3^, we could suppose that most of the formaldehyde loss in the chamber could be attributed to plant foliar uptake. A non-significant change in HCHO concentration in *C. comosum* leaves might suggest its effective metabolism by the plants. Moreover, even though the high concentration of HCHO was used, no visible harmful effects of this compound were detected in plant leaves. The results confirmed the previous findings that spider plants have a very strong capacity to remove formaldehyde from the air [2]. Moreover, Jian Li et al. [46] showed that *C. comosum* had the strongest resistance to formaldehyde as compared to other varieties of this species.

Nevertheless, HCHO did not appear to be neutral to the content and functioning of photosynthetic pigments. Chl a and Chl b are essential photosynthetic pigments determining the photosynthetic capacity of plants [33]. To optimize photosynthesis, plants dynamically change all these values (Chl a, Chl b, Chl a+b and Chl a/b ratio) depending on environmental conditions [47]. In the present studies, the HCHO application caused decreased contents of Chl b and Car as well as decreased emissions of Chl fluorescence after 48 h. These results are in line with other studies, which shows the negative influence of formaldehyde, especially at high concentrations, on the Chl b, and Car contents, and inhibition of photosynthesis [35]. Since Chl b plays the crucial role in the main antenna complex (LHCII) of photosystem II (PSII) [48,49], and there is a close relationship between Chl b content and LHCII, we may speculate that Chl b degradation induces LHCII degradation in *C. comosum* leaves, resulting in decreased emission of Chl fluorescence. 

The decrease in the Car content in HCHO-treated plants, observed in this work, might result from a decrease in the content of LHCII where Car participating in the xanthophyll cycle are peripherally associated with the LHCII [50]. Car are involved in the photoprotective mechanism of photosynthesis referred to as nonphotochemical quenching (NPQ). They are involved in quenching the excited states of Chl, scavenging ROS, and dissipating excess energy as heat [51,52]. Decreased Car contents result in the lack of chlorophyll-protecting molecules which made chlorophyll unstable and degraded [34,53]. Cheng et al. [54] in a study with tomato (*Solanum lycopersicum*) *ym* mutant (Yellow mutant) in which Car contents were significantly lower than in wild ones, showed that fifteen days after sowing, the Chl content in them began to decrease and it was accompanied by chloroplast degradation. The decreased Car content may have a significant influence on cell metabolism, including sugar and lipid metabolism, energy/metabolite transport, and redox systems in plant cells, as well as developmental, and environmental signals of the plant’s regulation under stress [34,55]. However, in the case of HCHO influence, their role needs to be clarified.

Interestingly, HCHO did not decreased the content of Chl a, which performs the primary function in photosynthesis [54]. Moreover, HCHO did not decrease total Chl content, which could suggest that upon HCHO treatment the conversion of Chl a to b was reduced. Similar trend was observed in plants grown under weak light, in which Chl b content decrease was related to significantly higher photosynthesis [54,56]. The ratio of Chl a/b could be a useful indicator of the efficacy of the photosynthesis and its increase is considered as the mechanism of plant acclimation to the changing environmental stressful conditions [54,57]. Therefore, it can be assumed that the increased ratio of Chl a/b observed by us may be related to the defense of *C. comosum,* to protect the burdened photosynthetic apparatus. Summing up, it is possible that changes in assimilation pigments observed by us may later lead to chlorosis as was described by other authors [54,58]. However, considering that HCHO at a high concentration did not inhibit all the parameters related to the functioning of photosynthetic pigments, we may suppose that it is related, in some extent, to the activation of plant defense responses, which should be further studied. 

The obtained data suggest that even a short exposure of *C. comosum* plants to HCHO-enriched air can influence carbohydrate metabolism in plant cells. *C. comosum* leaves responded to gaseous HCHO with considerable changes in soluble sugar contents. Reduction in the Suc content accompanied by an increase in the Gluc content may indicate disturbance in several processes related to carbon metabolism, similar to those observed in plants in anaerobic conditions. It was shown that the anaerobic conditions resulted in a significant decline in soluble carbohydrate content in roots and shoots of rice [59]. Our observations and those of others may suggest an increase in energy demand in both types of experimental conditions. Under optimal conditions, the conversion of Gluc to pyruvate in the cytosol relates to an initial input of ATP and production of the reduced NADH [60]. Upon hypoxia, respiratory ATP production decreases. To compensate for this, glycolytic activity increases and Gluc is utilized faster to produce ATP via the glycolytic pathway [61]. It is likely that, as in short-term hypoxia, in the early response to HCHO, plants must produce sufficient ATP and regenerate NADP^+^ and NAD^+^ which requires a greater supply of Gluc for glycolysis. Song et al. [25] showed in the study with labeled H^13^CHO that in *Arabidopsis thaliana* HCHO treatment enhanced the contents of Gluc and amino acids. It was indicated that it resulted from the assimilation of HCOOH by the Calvin cycle and the incorporation of ^13^C into the intermediates of the Calvin cycle and glycine via photorespiration [7,21]. 

The role of sugars synthesized in plants treated with HCHO has not been studied extensively. However, if we assume that the experimental conditions induced stress in the *C. comosum* cells, then changes in the content of soluble sugars can be considered as the result of the activation of defense responses. In this approach, the accumulated sugar in the form of Gluc, may play the role of carbon donor used for the biosynthesis of important primary as well as secondary metabolites [62]. Moreover, Gluc might affect photosynthesis, carbohydrate and lipid metabolism, osmotic homeostasis, nitrogen metabolism and protein synthesis, as well as gene expression, as it was presented in different plants, including these grown under unfavorable conditions [27,63,64]. On the other hand, the significant decrease in the Suc content might suggest that this sugar was intensively used for regulation of defense response to HCHO, and/or might be decomposed to Gluc, which in the given conditions is more useful to the *C. comosum* plant. Therefore, the role of Gluc and Suc and their transformations during plant response to HCHO should be studied in depth.

Even though amino acids are the large group of metabolites in which HCHO-derived carbon is incorporated, little is known about the enzymatic regulation of this process [25,26,28]. Plants contain multiple aminotransferase isoenzymes involved in amino acid synthesis and transformations, localized in various subcellular compartments including peroxisomes, mitochondria, and cytosol, expressed constitutively or in specific conditions [30,65,66]. In the present study, two enzymes involved in amino acid metabolism, AspAT and AlaAT, were investigated. An important role of aminotransferases in nitrogen metabolism in plants is related to their involvement in the synthesis of different amino acids from glutamate, and since aminotransferase-mediated reactions are reversible these enzymes may also participate in the formation of Glu by transferring the α-amino group from amino acids to 2-oxoglutarate, guaranteeing stability of Glu content in plant cells [67,68]. The results showed that 48 h after exposure to HCHO, the reduction of AspAT and AlaAT activities was observed in the *C. comosum* leaves. It was shown that AlaAT and AspAT changed their activities due to many stresses, e.g., aphid (*Sitobion avenae* F.) feeding on winter triticale seedlings [69], nickel-stressed wheat roots [41] or *A. thaliana* infection by *Pseudomonas syringae* [70]. In the present study, the decreased activity of AspAT may suggest disturbances in the generation of intermediates for the synthesis of several amino acids including aspartic acid (Asp) and Asn. Moreover, the diminished activity of AspAT limited the synthesis of Glu, which, in turn, might decrease AlaAT activity. 

Regarding the decrease in AlaAT activity, our results might suggest that the presence of HCHO shifted the balance of the reaction catalyzed by this enzyme from Glu synthesis in favor of Ala accumulation and reduced consumption of 2-oxoglutarate. It could be suggested that under HCHO conditions AlaAT might be responsible for maintaining a stable content of Ala or for its accumulation, and this enzyme is not involved in providing Glu. However, we studied only the total enzyme activity representing the activity of all enzyme isoforms, while AlaAT is present in plants in numerous molecular forms [71] and each of them might have responded differently to HCHO stress. Nevertheless, it is interesting that in plants treated with HCHO, the pathway of synthesis of Glu being a part of glutathione which participates in HCHO detoxication, is not enhanced. The mechanism of maintaining Glu content at the sufficient level to detoxify HCHO remains to be clarified.

Accumulation of phenolics in tissues is characteristic of plants which respond to various environmental stresses [38]. Plant phenolics confer various physiological functions for survival and adaptation to environmental disturbances. Regarding abiotic stresses, the beneficial role of phenolics results from their antioxidant properties as well as the ability to enhance plant structural barriers and defense signal transduction [38,72]. There are few studies in the literature describing the relationship between phenolics and formaldehyde. It is known that polyphenols can react with formaldehyde [73], and phenols contained in plant extracts or seeds reduce formaldehyde concentration [74,75]. However, there are no studies on the effect of plant exposure to exogenous HCHO on qualitative and quantitative changes in the pool of phenols in the plant cell. This study showed no significant changes in the total concentration of phenolics in the leaves of the *C. comosum* after exposure to HCHO. We suppose that 48 h of HCHO treatment could be too short to cause changes in the metabolism of phenolics, as suggested in other studies [38]. However, taking into account the results of other studies, which showed that HCHO influences lignin synthesis and enhances expression of enzymes that are involved in phenolic transformations, such as peroxidase POD [11,12], we believe that a deeper study of the synthesis and identification of selected classes of phenolics should be performed.

## 5. Conclusions

In summary, in the present study, the ability of *C. comosum* plants to effectively remove HCHO from the air, even at unrealistically high concentrations of 20 mg m^−3^, within two days, was confirmed. Even though a high concentration of HCHO was used, no visible harmful effects of this compound were detected in plant leaves and absorbed HCHO seemed to be effectively metabolized by *C. comosum* cells. In the present study, the HCHO application caused no changes in total Chl and Chl a content; increased Chl a/b ratio, and decreased Chl b and Car content, as well as Chl fluorescence after 48 h, suggesting that the volatile compound might have a negative impact on photosynthetic pigments, but *C. comosum* might have rearranged its synthesis and functioning towards the protection of the plant photosynthetic apparatus from HCHO-induced damages. A decrease in Suc accompanied by an increase in Gluc suggests that these sugars are extensively affected in HCHO-treated *C. comosum* metabolism, towards the utilization in metabolism of Suc and synthesis or accumulation of Gluc. The results showed that HCHO treatment decreased activities of AspAT and AlaAT, suggesting that these enzymes do not play any pivotal role in amino acid transformations during HCHO assimilation. Short, 48 h treatment with HCHO did not influence phenolic metabolism. In our evaluation, HCHO-induced changes in leaf tissues resemble those observed in the tissues of plants growing under hypoxic conditions, and to some extent those triggered by photorespiration. However, a more detailed study is needed. Similarly, the problem of whether the observed changes are adaptive or the beginning of stress-related pathological changes and the beginning of defense response to HCHO also needs clarification.

The presented results encourage further studies of (i) the photosynthetic pigment transformations and functioning of photosynthetic apparatus, including antenna complex (LHCII), (ii) determination of the synthesis of the other sugars, in order to check their quantitative and qualitative relations, (iii) studies of other enzymes that may be involved in the synthesis of amino acids and phenolic compounds, and (iv) deeper study of the synthesis and identification of selected classes of phenolics that may be involved in the defense responses of *C. comosum* to high concentrations of HCHO. 

These studies may be an important step in elucidating the properties of *C. comosum* that allow it to effectively remove HCHO from the air. Studies with HCHO in high concentrations may reveal practical knowledge enabling more efficient use of plants in places burdened with the presence of HCHO in the air in high concentrations, such as factories that make paints, furniture, and synthetic materials.

## Figures and Tables

**Figure 1 cells-12-00232-f001:**
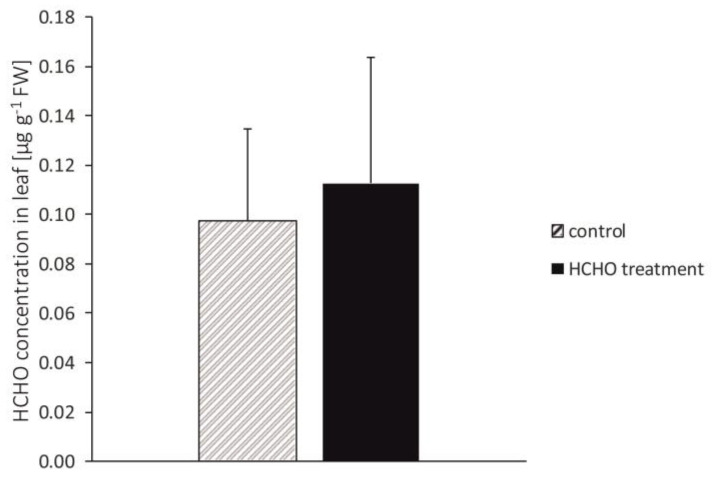
Effect of treatment with gaseous formaldehyde applied as 20 mg m^−3^ on formaldehyde concentration in *C. comosum* leaf tissues after 48 h of plant exposure. The data are means ± SD.

**Figure 2 cells-12-00232-f002:**
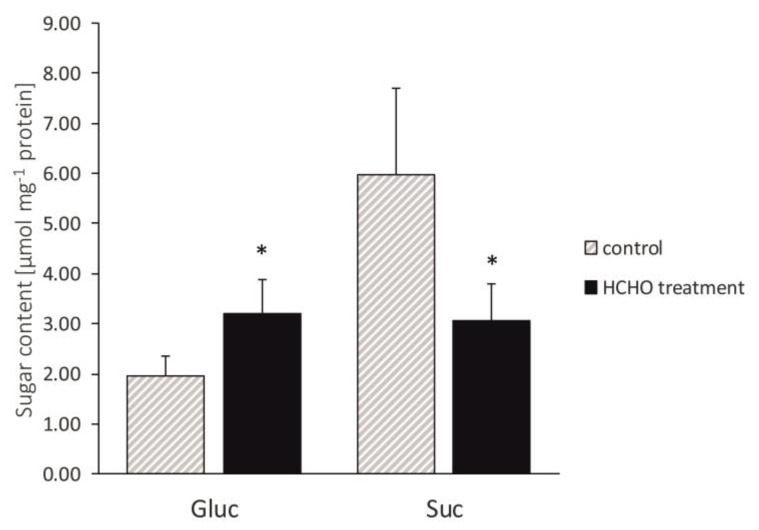
Effect of treatment with gaseous formaldehyde applied as 20 mg m^−3^ on Gluc and Suc contents in *C. comosum* leaf tissues after 48 h of plant exposure. Data are means ± SD. * indicates values that differ significantly from the control at *p* < 0.05.

**Figure 3 cells-12-00232-f003:**
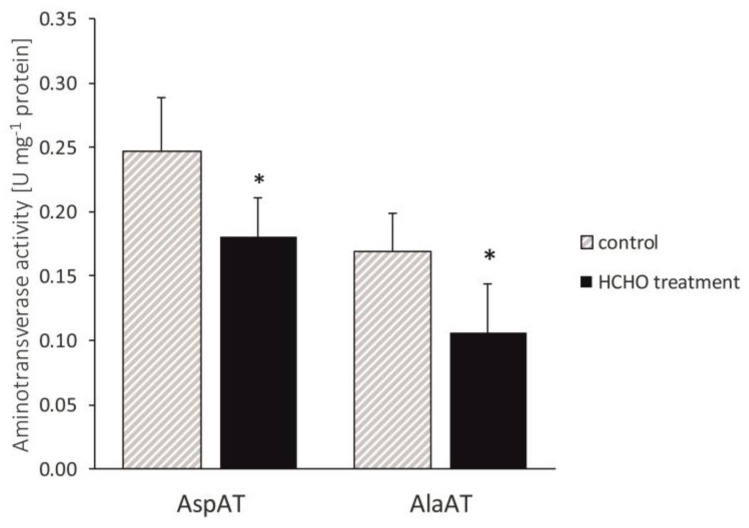
Effect of treatment with gaseous formaldehyde applied as 20 mg m^−3^ on AspAT and AlaAT activities in *C. comosum* leaf tissues after the 48 h of plant exposure. The data are means ± SD. * indicates values that differ significantly from the control at *p* < 0.05.

**Figure 4 cells-12-00232-f004:**
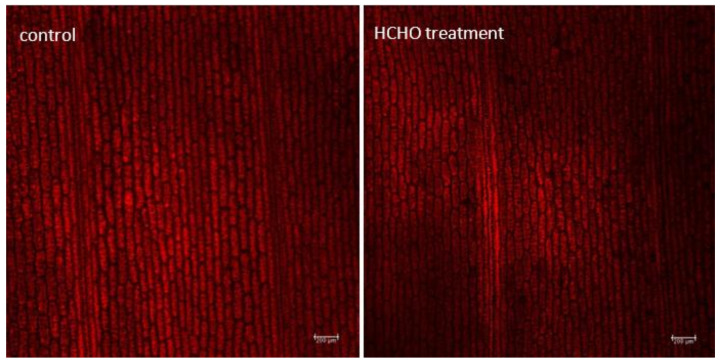
Effect of treatment with gaseous formaldehyde applied in a dose of 20 mg m^−3^ on the intensity of chlorophyll fluorescence in *C. comosum* leaf tissue after 48 h of plant exposure.

**Figure 5 cells-12-00232-f005:**
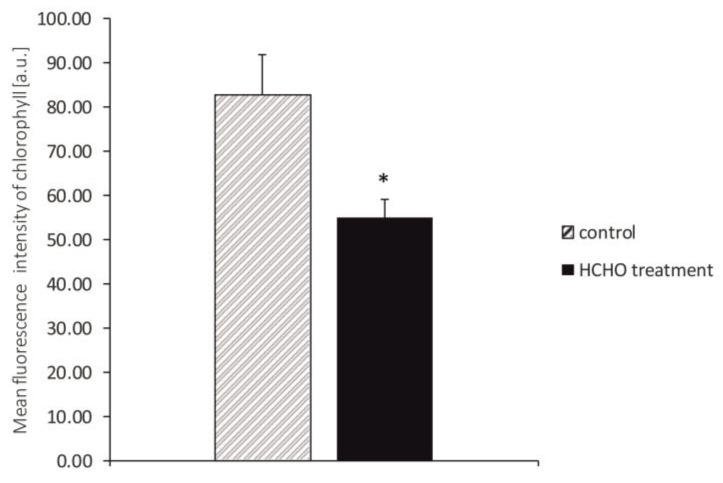
Effect of treatment with gaseous formaldehyde applied as 20 mg m^−3^ on intensity of chlorophyll fluorescence in *C. comosum* leaf tissues after the 48 h of plant exposure. * indicates values that differ significantly from the control at *p* < 0.05.

**Figure 6 cells-12-00232-f006:**
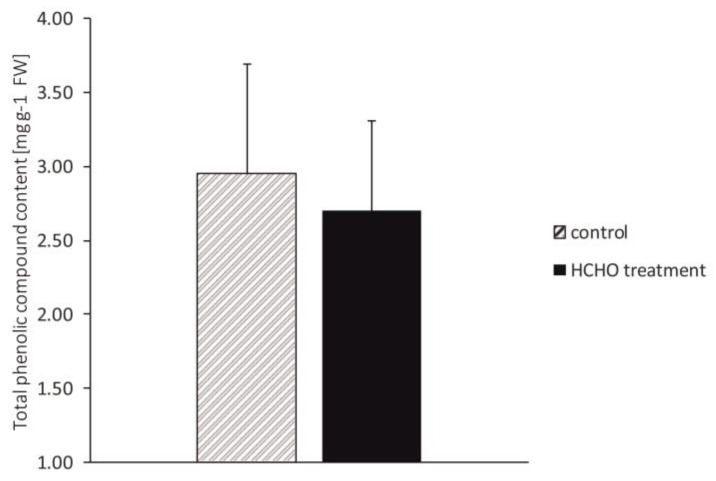
Effect of treatment with gaseous formaldehyde applied as 20 mg m^−3^ on total phenolic content in *C. comosum* leaf tissues after 48 h of plant exposure. The data are means ± SD.

**Table 1 cells-12-00232-t001:** Effect of treatment with gaseous formaldehyde applied as 20 mg m^−3^ on photosynthetic pigment content: Chl a, Chl b, Chl a+b, and Car, as well as Chl a to Chl b ratio in *C. comosum* leaf tissues after 48 h of plant exposure.

	Chl a (µg g^−1^FW)	Chl b (µg g^−1^FW)	Chl a+b (µg g^−1^FW)	Chl a/b ratio	Car (µg g^−1^FW)
Control	461.29 ± 92.58	308.56 ± 54.02	806.30 ± 101.86	1.64 ± 0.48	111.29 ±18.36
HCHO	438.71 ± 96.46	248.86 ± 70.71 *	729.52 ± 104.48	2.14 ± 0.56 *	70.71 ± 24.28 *

The data are means ± SD. * indicates values that differ significantly from the control at *p* < 0.05.

## Data Availability

Data are contained within the article.

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
