# Peer review of "New Insight into Short Time Exogenous Formaldehyde Application Mediated Changes in Chlorophytum comosum L. (Spider Plant) Cellular Metabolism"

_cells, 2023, doi:10.3390/cells12020232_

Round 1
Reviewer 1 Report
I consider that the topic of the article is not suitable for the profile of the journal. The results only partially describe the effects caused by exogenous exposure to HCHO and the results are not explained in depth.
Since the Journal Cells profile is cellular in nature, the research question should be based on these terms, the same should be for the discussion section.

Author Response
Dear Reviewer,
Thank you for your valuable comments. We have considered all of them. Once again, we critically looked at the manuscript. We added some data which show the response of C. comosum to the formaldehyde treatment, at the cellular level, mainly related to the imaging of chlorophyll autofluorescence in the cells. We changed the Introduction and Discussion sections, to be more related to the results and not digress from the subject. We rewrote conclusions and added future perspectives of studies. Thanks to your valuable comments our manuscript could be modified and improved. We hope that in the present version the manuscript is more suitable for the profile of the journal.
We responded to the comments in the attached PDF file (the previous version of the manuscript). All the changes were made in the present version of the manuscript.
Kind regards,
M.Eng. Urszula Świercz-Pietrasiak

Reviewer 2 Report
The paper discusses the biochemical changes in C. comosum leaves induced by 48 h exposition to exogenous HCHO applied as 20 mg m-3.
There are many major points should be clarified before considering its publication
· The introduction is too long and it is better that the authors concentrate more on the formaldehyde metabolism and its phytoremediation with plants as the authors show what the literature reached and the limits in their studies and then they introduce their work novelty.
Some references can be useful
- A review of plants formaldehyde metabolism: Implications for hazardous emissions and phytoremediation. https://doi.org/10.1016/j.jhazmat.2022.129304.
- Assimilation and metabolism of formaldehyde by leaves appear unlikely to be of value for indoor air purification. https://doi.org/10.1046/j.1469-8137.2000.00701.x.
- Absorption and metabolism of formaldehyde in solutions by detached banana leaves. https://doi.org/10.1016/j.jbiosc.2013.10.017.
- Tolerance of fifteen hydroponic ornamental plant species to formaldehyde stress. https://doi.org/10.1016/j.envpol.2020.115003.
- A novel formaldehyde metabolic pathway plays an important role during formaldehyde metabolism and detoxification in tobacco leaves under liquid formaldehyde stress. https://doi.org/10.1016/j.plaphy.2016.04.028
- C1 metabolism plays an important role during formaldehyde metabolism and detoxification in petunia under liquid HCHO stress. https://doi.org/10.1016/j.plaphy.2014.08.017
- Studies for Absorption of Formaldehyde by Using Foliage on Wild Tomato Species. https://doi.org/10.2503/hortj.OKD-070.
- Co-overexpression of AtSHMT1 and AtFDH induces sugar synthesis and enhances the role of original pathways during formaldehyde metabolism in tobacco. https://doi.org/10.1016/j.plantsci.2021.110829.
- Formaldehyde biofiltration as affected by spider plant. https://doi.org/10.1016/j.biortech.2010.03.128.
· Line 149-162In the last part of the introduction the aim of the study is not apparent it should be rewritten in another way to be clearer. Especially the section about the paper's novelty; the authors should concentrate on the metabolic pathways they studied in their work.
· Line 184 In the materials and method section at 2.1. plant material part, the authors said that “The concentration was chosen based on our preliminary studies” if this is a previous study add its reference or if not add the results of these preliminary studies as supplementary data
· Line 190: “The temperature, relative humidity, and photoperiod in the chamber were controlled and kept the same as during the initial plant breeding.” Specify the temperature, relative humidity, and photoperiod in the study
· Line 192-198: “2.2. Treatment of plants with formaldehyde” this is discussed before in the previous section, you add the detailed information about formaldehyde treatment to the previous section and change this title for example change to “Samples preparation”
· Line 237- 238: Determination of the photosynthetic pigments section “To determine the contents of Chl a, Chl b, and Car, the frozen leaves were extracted three times with 80% acetone (1:20 w/v) and centrifuged (33,000 x g, 15 min).” How do you determine the pigments in frozen samples? The pigments determination must be performed on fresh leaves not frozen and the determination should be performed in the dark as the pigments are rapidly broken by light or in the frozen storage. This point should be considered as this may cause distortion to the pigments results.
· Line 248-249: The unit used for glucose and sucrose determination is “μg per mg protein determined in the extract used for enzyme activity analyses”. clarify this point.
· Line 257-258: the unit for phenolic compounds used as “milligrams per gram of dried extract”. How you calculated it in the dried extract? You said that you used the supernatant of the ground frozen leaves. Clarify this point
· Line 297-298: in Table (1) title, “The data are means ± SD. * indicates values that differ significantly from the control at P<0.05” this sentence should not be in the table title. It is added as table footnotes. Also, in the table there is not any *, the authors add P<0.05 at the significantly different values, and replace this with the * as you mentioned.
· Line 320-321: “The results confirmed the previous findings that Spider plants have a very strong removal capacity of formaldehyde from the air” define how the result of the present study confirmed the formaldehyde removal and you did not measure the formaldehyde in the box after at the end of the experiment and there was a non-significant change in its content in the plants.
· The whole discussion is missy and should be reorganized so that the authors can begin with changes in the pigments then, the carbon metabolism represented in the glucose and sucrose content and then the nitrogen metabolism changes.
· The conclusion is too long and not clear it should be rewritten. Concentrate on the important results and then write your future perspective.
·
Author Response
Dear Reviewer,
Thank you for your valuable comments. We have considered all of them. Once again, we critically looked at the manuscript. We added some data which show the response of C. comosum to the formaldehyde treatment, at the cellular level, mainly related to the imaging of chlorophyll autofluorescence in the cells. We changed the Introduction and Discussion sections, to be more related to the results and not digress from the subject. We rewrote conclusions and added future perspectives of studies. Thanks to your valuable comments our manuscript could be modified and improved. We hope that in the present version the manuscript is more suitable for the profile of the journal.
We responded to the comments in the attached, Word file. All the changes were made in the present version of the manuscript.
Kind regards,
M.Eng. Urszula Świercz-Pietrasiak
Round 2
Reviewer 1 Report
I consider that paper is aceptable in present form. The authors were realized the modifications appropiate.
Reviewer 2 Report
The authors performed the required revision carefully. The manuscript can be published after minor revision to the manuscript English typing errors